

# Estrogen receptor—positive breast cancer survival prediction and analysis of resistance–related genes introduction

Chen Shuai[1], Fengyan Yuan[2], Yu Liu[3], Chengchen Wang[3], Jiansong Wang[3] and Hongye He[4]

[1] Department of Breast and Thyroid Surgery, Yiyang Central Hospital, Yiyang, Hunan, China
[2] Hunan Normal University of Medicine, Changsha, Hunan, China
[3] Hunan Provincial People's Hospital, Changsha, Hunan, China
[4] Department of General Surgery, The Second Xiangya Hospital, Central South University, Changsha, Hunan, China

## ABSTRACT

**Background**. In recent years, ER+ and HER2- breast cancer of adjuvant therapy has made great progress, including chemotherapy and endocrine therapy. We found that the responsiveness of breast cancer treatment was related to the prognosis of patients. However, reliable prognostic signatures based on ER+ and HER2- breast cancer and drug resistance-related prognostic markers have not been well confirmed, This study in amied to establish a drug resistance-related gene signature for risk stratification in ER+ and HER2- breast cancer.

**Methods**. We used the data from The Cancer Genoma Atlas (TCGA) breast cancer dataset and gene expression database (Gene Expression Omnibus, GEO), constructed a risk profile based on four drug resistance-related genes, and developed a nomogram to predict the survival of patients with I-III ER+ and HER2- breast cancer. At the same time, we analyzed the relationship between immune infiltration and the expression of these four genes or risk groups.

**Results**. Four drug resistance genes (AMIGO2, LGALS3BP, SCUBE2 and WLS) were found to be promising tools for ER+ and HER2- breast cancer risk stratification. Then, the nomogram, which combines genetic characteristics with known risk factors, produced better performance and net benefits in calibration and decision curve analysis. Similar results were validated in three separate GEO cohorts. All of these results showed that the model can be used as a prognostic classifier for clinical decision-making, individual prediction and treatment, as well as follow-up.

# INTRODUCTION

Breast cancer is a common diagnosed malignant tumor that lead to cancer-related mortality in women worldwide; it can mainly classified into invasive ductal carcinoma (IDC) and invasive lobular carcinoma (ILC) by histopathological classification (*Dianatinasab et al., 2019*; *Li, Uribe & Daling, 2005*; *Viale, 2012*). The basic status of the estrogen receptor (ER), progesterone receptor (PR), and human epidermal growth factor receptor 2 (HER2)

Corresponding author
Hongye He, hhyhongye@csu.edu.cn, qingyunshuimo@sina.com

play a crucial role in molecular subtypes of breast cancer. Previous molecular studies demonstrated that breast cancer can be classified into Luminal A (ER+/PR+/HER2-/low Ki-67); Luminal B (ER+/PR+/HER2-/+/high Ki-67), HER2-overexpression (ER-/PR-/HER2+) and triple negative breast cancers (ER-/PR-/HER2-) (*Al-Thoubaity, 2020*; *Goldhirsch et al., 2011*). Different molecular subtypes exhibit different responsiveness to therapies. For instance, most patients of breast cancer are present as ER+ and HER2- and can be benefit from endocrine therapy.

Although surgery-based comprehensive therapy can significantly improve the prognosis of breast cancer, the endocrine resistance may occur in approximately 33% of patients with endocrine therapy (*Achinger-Kawecka et al., 2020*). When endocrine-resistance appears, chemotherapy is needed to control the disease progression (*Zhu et al., 2018*). Nevertheless, a small subset of patients are resistant to both endocrine therapy and chemotherapy, so that lead to poor prognosis. In clinical practice, tamoxifen is one of the first-line endocrine therapies that competes with estrogen for ER and counteracts the estrogen signal pathway, which retards breast cancer growth (*Swaby, Sharma & Jordan, 2007*; *Ye et al., 2019*). Similarly, Doxorubicin is the most widely used DNA-damaging chemotherapeutic drug (*Pfitzer et al., 2019*). Many studies have been investigated to explore the mechanism of drug resistance. Several pathways including the PI3K/Akt/mTOR pathway, fibroblast growth factor receptor pathway, etc. are activated to reverse the growth suppression induced by tamoxifen and doxorubicin (*Guo et al., 2016*; *Li et al., 2020a*; *Li et al., 2019*; *Lu et al., 2019*; *Shah et al., 2015*; *Turner et al., 2010*). In addition, several studies showed that tumor infiltrating lymphocytes (TILs) play a pivotal role in chemotherapy response and associate with clinical outcome in all subtypes of breast cancer (*Denkert et al., 2010*; *Guo et al., 2016*; *Stanton & Disis, 2016*; *Turner et al., 2010*).

The clinicopathological risk factors, such as TNM stage and histological grade, are usually used to assess the survival risk of breast cancer. However, as a result of molecular differences, clinical prognoses are mainly different even in patients with histologically similar tumors, which indicates that only using these risk factors as evaluation indicators cannot accurately predict the prognoses of breast cancer patients (*Tang et al., 2019*). Therefore, it is necessary to identify novel prognostic biomarkers to improve the prognosis evaluation ability of breast cancer.

In this study, we used the data obtained from The Cancer Genome Atlas (TCGA) breast cancer dataset and the Gene Expression Omnibus (GEO) database, then built a four resistance-related genes based risk signature and developed a nomogram to predict the survival of patients with ER+ and HER2- breast cancer in stage I-III. Finally, we analyzed the association between immune infiltrates and the four genes' expression or risk groups.

## MATERIALS & METHODS

### Data download

The raw counts of the RNA sequence data of the TCGA-BRCA dataset were downloaded from the TCGA website (https://portal.gdc.cancer.gov/). The clinical features and the disease-free survival (DFS) status of these patients were downloaded from the cBioportal

**Table 1  Baseline data sheet for patients in TCGA ($n = 473$).**

|  | NO. | % |
|---|---|---|
| Age |  |  |
| $\geq$50 year | 348 | 73.57% |
| <50 year | 125 | 26.43% |
| T |  |  |
| T1 | 139 | 29.39% |
| T2 | 259 | 54.76% |
| T3 | 69 | 14.59% |
| T4 | 6 | 1.27% |
| N |  |  |
| N0 | 214 | 45.24% |
| N1 | 168 | 35.52% |
| N2 | 53 | 11.21% |
| N3 | 35 | 7.40% |
| NX | 3 | 0.63% |
| Pathologic Stage |  |  |
| Stage I | 95 | 20.08% |
| Stage II | 265 | 56.03% |
| Stage III | 113 | 23.89% |
| PR Status |  |  |
| positive | 411 | 86.89% |
| negative | 60 | 12.68% |
| unknown | 2 | 0.42% |

database (http://www.cbioportal.org). We selected 473 ER+ and HER2- breast cancer patients in stage I-III or the subsequent analysis (Table 1). The R software (version 3. 6. 3) was used to standardize and process data.

GSE21653 (100 samples of ER+ and HER2- early invasive breast cancer), GSE17705 (298 samples of ER+ breast cancer) and GSE11121 (200 patients) datasets were acquired from GEO database (https://www.ncbi.nlm.nih.gov/geo/) as the validation cohorts. GSE21653 has DFS as the endpoint, GSE17705 has relapse-free survival (RFS) as the endpoint and GSE11121 has distant metastases-free survival (DMFS) as the endpoint.

## Screening of differentially expressed genes

The resistance-related differentially expressed genes (DEGs) were detected with the "limma" package in R software from GSE67916 (DEGs between MCF7 tamoxifen resistant (MCF7/TamR) cell lines and MCF7 parental cell lines) and GSE24460 (DEGs between MCF7 doxorubicin resistant (MCF/ADR) cell lines and MCF7 parental cell lines). Defined genes with adjusted $p$ values <0. 05 and |log2foldchange| values >1. 5 as DEGs. The genes both up-regulated (16 genes) and down-regulated (21 genes) in the two GEO datasets were finally screened (Fig. S1B). Volcano plots and heatmaps were performed using the R packages "ggpubr", "ggthemes" and "pheatmap" to exhibit the differentially expressed genes (Figs. S1A, Fig. 1C).

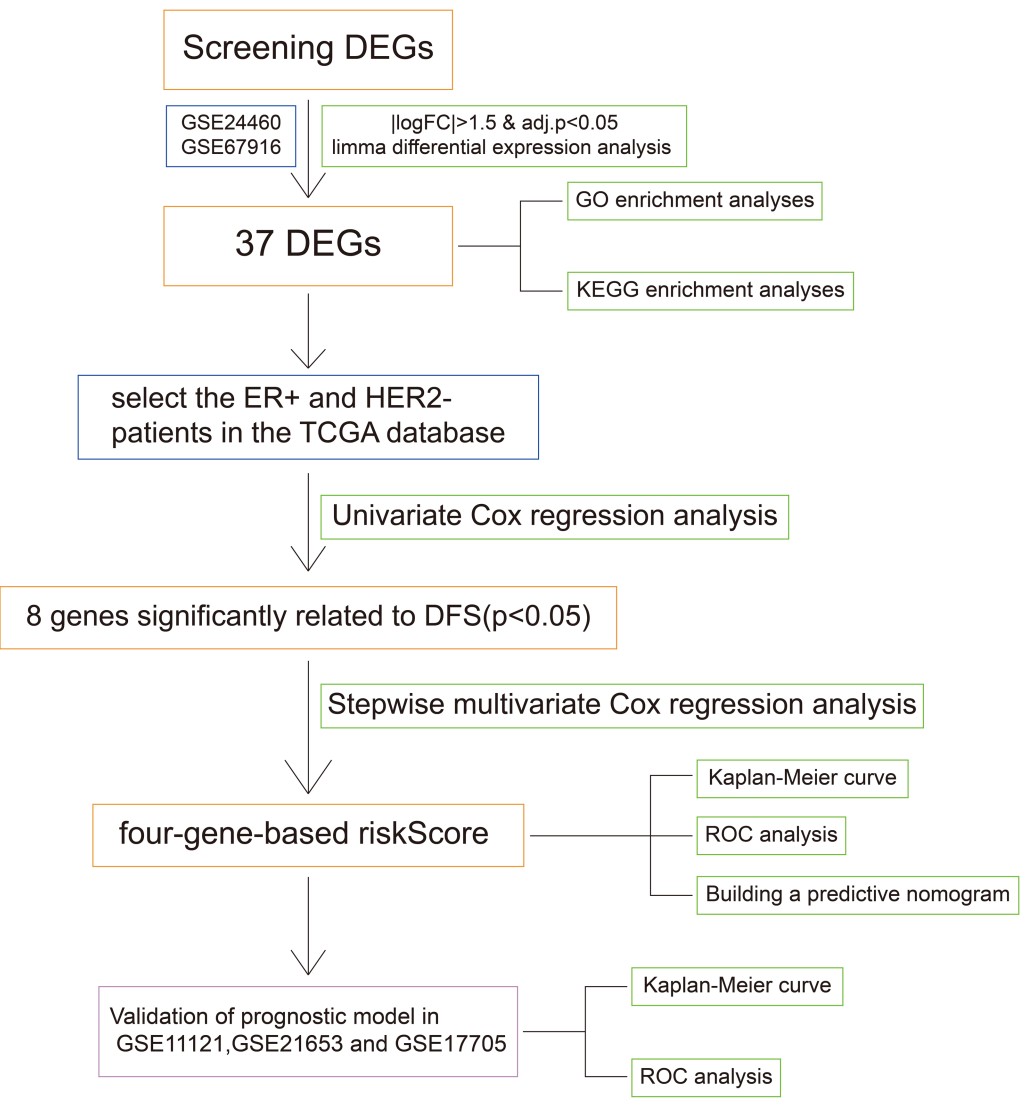

**Figure 1  The workflow of the identification of breast cancer resistance-related four gene signatures.**

## Identification and validation of a risk signature associated with breast cancer

The data downloaded from TCGA datasets were used as the training cohort to analyze the correlation of genes' expression with the DFS time based on univariate Cox regression analysis. Patients without survival for less than 0 days were removed to identify prognostic genes, only the genes with the $p$ values < 0.05 were selected as candidates (Fig. 2A). Then, the candidate genes were selected for stepwise multivariate Cox regression analysis (Fig. 2B). The Akaike information criterion (AIC) was used to avoid over-fitting, we chosen the best-fit predictive model with the lowest AIC (AIC =359. 44). Finally, a four-gene risk signature was established and calculated their risk coefficients use the formula: The riskScore = (−0. 175*expression of AMIGO2) + (−0. 210* expression of

LGALS3BP) + (−0. 106* expression of SCUBE2) + (−0. 293* expression of WLS). Using the optimal riskScore cut-off value (−10.09) calculated by the R package "survminer", we divided patients in both the training cohort and the validation cohort into high-risk and low-risk group. Then, Kaplan–Meier curve and log-rank test were used to analyze the different survival time between high-risk group and low-risk group. Subsequently, a time-dependent receiver operating characteristic (ROC) curve was calculated to compare the sensitivity and specificity of risk factors.

### Development of the prognostic nomogram

To identify independent predictors of DFS, we tested clinical risk factors and the gene-based riskScore. Then we used the R package "rms" to establish the prognostic nomogram. The discrimination and calibration abilities of the nomogram were assessed with a concordance index (C-index) and calibration plot. For further evaluation of the predictive performance, we also used the time-dependent ROC curves.

### Function and pathway enrichment analyses

The R package "clusterProfiler" was used to classify the 37 DEGs' GO terms or KEGG pathways. Functional enrichment analyses were carried out for GO terms and KEGG pathways through the significance threshold of $p$ value < 0. 05.

### Immune infiltration analyses

The 22 kinds of immune cell infiltration levels of TCGA-BRCA dataset were quantified by CIBERSORT algorithm with 1,000 permutations (http://cibersort.stanford.edu/) (*Chen et al., 2018a*). CIBERSORT calculates a $p$ value to confirm the accuracy of the results, only the samples with the $p$ value < 0.05 were selected for analyzing the immune infiltration levels between the high- and low- risk groups. Moreover, to explore the correlation between the four genes' expression and the abundance of immune cell infiltrates (B cells, CD4+ T cells, CD8+ T cells, neutrophils, macrophages, and dendritic cells), the Tumor Immune Estimation Resource (TIMER, https://cistrome.shinyapps.io/timer/) was utilized.

### Statistical analyses

All the statistical analyses were performed with R software (version 3. 6. 3). All the statistical analyses included univariate and multivariate Cox regression analysis, ROC curve analysis, Kaplan–Meier survival analysis and correlation analysis. The correlation analysis was based on the R package "corrplot". Statistical differences were compared by Wilcoxon test between two groups and Kruskal-Wallis H test for multigroup comparison. $P < 0.05$ was considered statistically significant.

## RESULTS

### Screening DFS-related resistance-related DEGs

The flow chart was shown in Fig. 1. To screen the DFS-related resistance-related DEGs, two datasets was downloaded from GEO database. After "limma" filtering between MCF7 doxorubicin or tamoxifen resistant cell lines and MCF parental cell lines (Fig. S1A), the intersection of up-regulated and down-regulated genes was taken respectively (Fig. S1B).

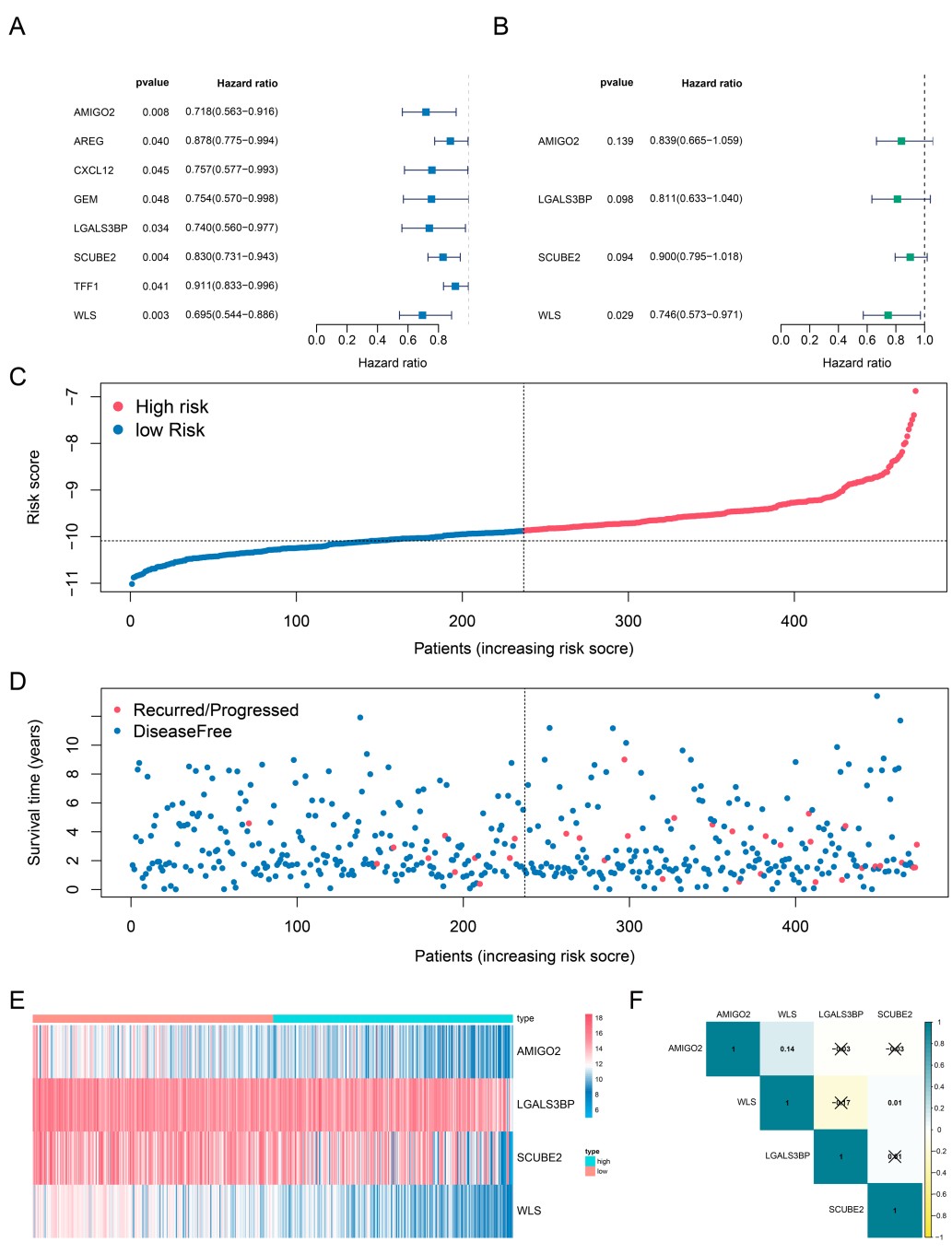

**Figure 2 Determination and analysis of the eight genes signature in the training cohort.** (A, B) Univariate Cox (A) and multivariate Cox (B) regression analysis results. (C) The four-gene signature riskScore distribution plot. (D) The vital status plot of patients. (E) The heatmap of the expression profiles of the four genes. (F) The correlation of four genes in TCGA.

A total of 37 resistance-related DEGs were obtained. The expression profile of the 37 DEGs in GSE24460 and GSE67916 were shown in Fig. S1C.

Next, we investigated whether genes could be used as prognostic candidates in ER positive (ER+) and HER2 negative (HER2-) breast cancer patients in stage I-III. The univariate Cox regression analysis found that eight genes were associated with DFS time in ER+ and HER2- breast cancer, all of these genes were protective factors (Fig. 2A). The eight candidates were further analyzed by multivariate Cox regression analysis and examined four genes (*AMIGO2, LGALS3BP, SCUBE2* and *WLS*) as the prognostic candidates (Table 2, Fig. 2B). In addition, we verified the prognosis of these four genes in the Kaplan–Meier Plotter website (https://kmplot.com/analysis/). Consistent with our results, *WLS* and *SCUBE2* were protective factors, high expression of the two genes had longer RFS time (Fig. S2C). The results of other two genes were not statistically significant. A risk score formula (the formula shown in methods) derived from the expression levels of the four prognostic candidates was constructed by the multivariate Cox regression analysis.

## Identification and validation of a four-gene based risk signature

All enrolled patients were divided into high- and low-risk groups with its optimal cut-off value among the training cohort by using the riskScore. The survival status plot and heatmap for the expression of the four genes were showed in Figs. 2C–2D. The prognostic capacity of the four-gene based riskScore was estimated by calculating the AUC of a time-dependent ROC curve (Fig. 3A). The AUCs of the four-gene risk signature were 0.686 and 0.714 for 3- and 5-year DFS respectively, suggesting that the risk signature had certain sensitivity and specificity. Kaplan–Meier curve showed that patients in the high-risk group had a lower DFS time than those in the low-risk group (Fig. 3A).

We also analyzed the correlation of the four genes in ER+ and HER2- breast cancer patients in TCGA-BRCA dataset. The correlation heatmap showed that the maximum R values was 0.14, indicating that these four genes were independent of each other (Fig. 2F).

To validate the prognostic prediction performance of riskScore, the same as training cohort, patients in the validation cohorts were classified into high- and low-risk groups using the same formula obtained from the training cohort. In all validation cohorts, patients with a high riskScore had a shorter survival time and a high riskScore was associated with worse prognosis (Figs. 3B–3D). Additionally, the time-dependent ROC analyses for the survival prediction of the risk signature showed AUCs of 0. 624 at 3 years, 0. 613 at 5 years in GSE21653 and 0. 618 at 3 years, 0. 616 at 5 years in GSE17705, demonstrating that this risk signature was capable of predicting survival in ER+ and HER2- or ER+ and HER2 equivocal patients (Figs. 3B–3C). Moreover, we also used the time-dependent ROC analysis to explore the prognostic prediction performance if the subtype of breast cancer was unknown. The AUCs were 0.730 at 3 years and 0.669 at 5 years in GSE11121 (Fig. 3D), suggesting that our risk signature also has good predictive ability when breast cancer subtype is unknown.

## Stratified analyses with clinical factors

As shown in Fig. 4, the four-gene based risk signature could serve as a promising candidate for predicting the survival of breast cancer in different subtypes stratified by the clinical

**Table 2  Four resistance-related genes associated with DFS.**

| Gene | Coef | HR | Group |
|------|------|-----|-------|
| AMIGO2 | −0.17541 | 0.839117 | Down-regulated |
| LGALS3BP | −0.20954 | 0.810961 | Up-regulated |
| SCUBE2 | −0.10588 | 0.899535 | Down-regulated |
| WLS | −0.29345 | 0.745685 | Up-regulated |

factors, including N0-1 type ($p = 0.0014$), N2-3 ($p = 0.026$), T1-2 ($p = 0.0006$) T3-4 ($p = 0.04$), stage I-II ($p = 0.0022$), stage III ($p = 0.0078$), PR positive (PR+, $p < 0.0001$) patients, respectively. But the PR negative (PR-) subtype was not statistically significant. As shown in Fig. 5, the riskScore value was higher expression in PR- subtype than in PR+ subtype, *SCUBE2* and *WLS* were higher expression in PR+ subtype than in PR-subtype, the expression of *SCUBE2* was negatively correlated with T and Stage, these results were consistent with the clinical prognostic significance.

## Functional enrichment analyses

To explain the function of the resistance-related DEGs, we performed enrichment analyses of the GO terms and KEGG pathways (Figs. S2A–S2B).

GO analyses showed that these resistance-related DEGs were enriched in several biological processes (BP), including antigen processing and presentation of peptide antigen *via* MHC class I, antigen processing and presentation of exogenous peptide antigen *via* MHC class I and epithelial structure maintenance (Fig. S2A).

KEGG analysis showed that the resistance-related DEGs were related to endocytosis, cellular senescence, antigen processing and presentation and adherens junction (Fig. S2B).

## Revealing the correlation between the risk signature and tumor-infiltrating immune cells

Because TILs play a key role in chemotherapy response and our enrichment analyses results showed that the DEGs associated with antigen presentation, first, we evaluated the abundance of the 22 kinds of immune cells in each TCGA-BRCA patient (Fig. 6A). Next, we utilized the Wilcoxon test to compare the difference immune cell infiltration levels between the high- and low groups and found that B cells memory ($p = 0.042$) and T cells follicular helper ($p = 0.016$) were high infiltrated in the high-risk group, and B cells naive ($p = 0.047$) and mast cells resting ($p = 0.042$) were low infiltrated in the high-risk group (Fig. 6B). Unfortunately, due to the limitation of sample size, the relationship between the immune cell infiltration levels and survival is unknown. Whereafter, we assessed the correlation between the four resistance-related DEGs and six infiltrating immune cells by using the TIMER database. The results revealed that the genes had good correlation with immune cells (Fig. S3).

## Building a prognostic nomogram combining riskScore with clinical risk factors

Nomograms are widely used in predicting cancer patients' prognoses, principally because they have the ability to decrease the statistical prediction models into a single numerical

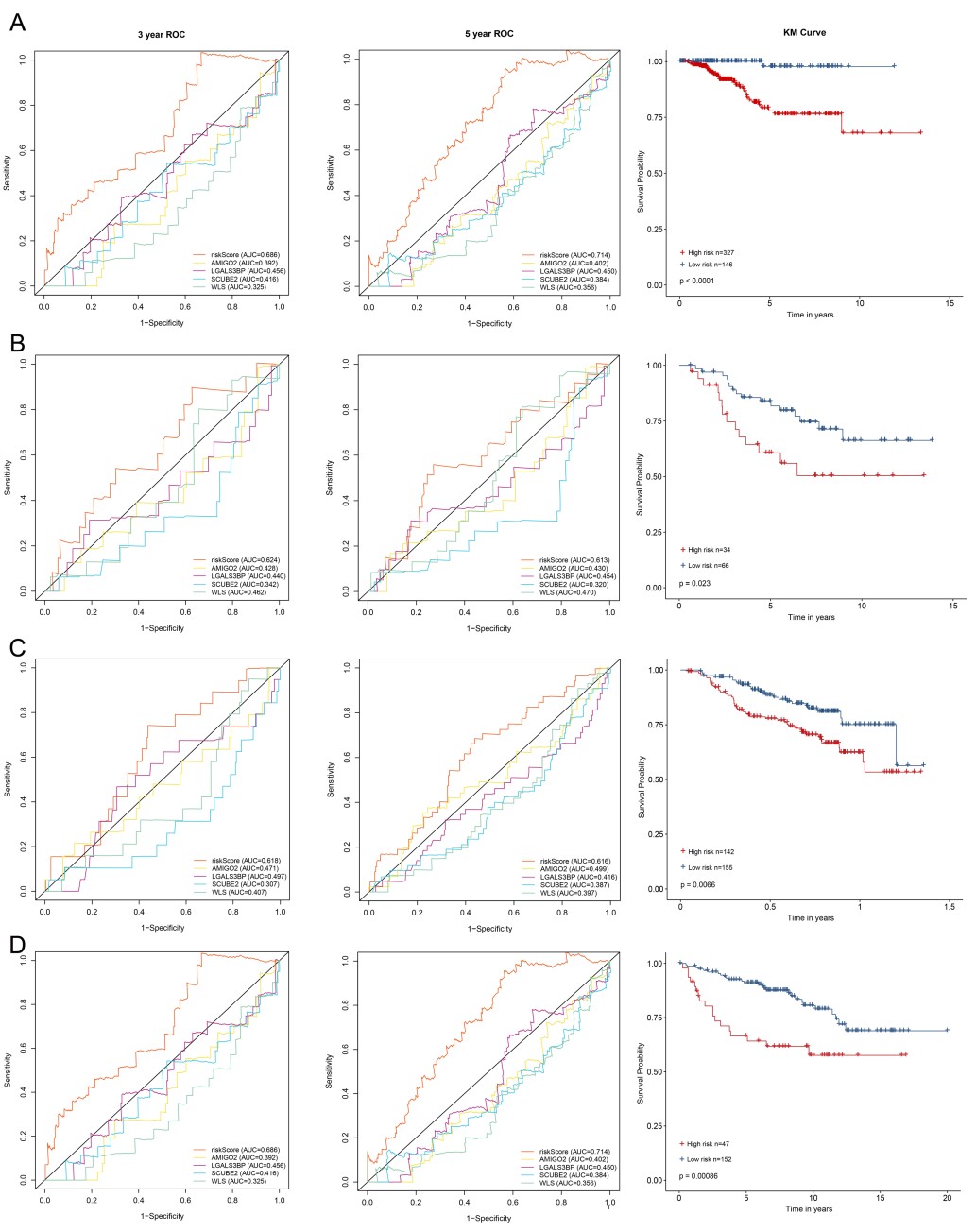

**Figure 3** Kaplan–Meier survival curves of DFS between high-risk and low-risk patients and ROC curves in (A) TCGA-BRCA, (B) GSE21653, (C) GSE17705 and (D) GSE11121.

assessment, which is tailored to the profile of an individual patient (*Iasonos et al., 2008*; *Long et al., 2018*).

In this article, we combined the four-gene based riskScore with clinical risk factors to construct a nomogram to assess the probability of 3- and 5- year DFS for patients with ER+ and HER2- early stage invasive breast cancer in TCGA-BRCA dataset. After univariate and

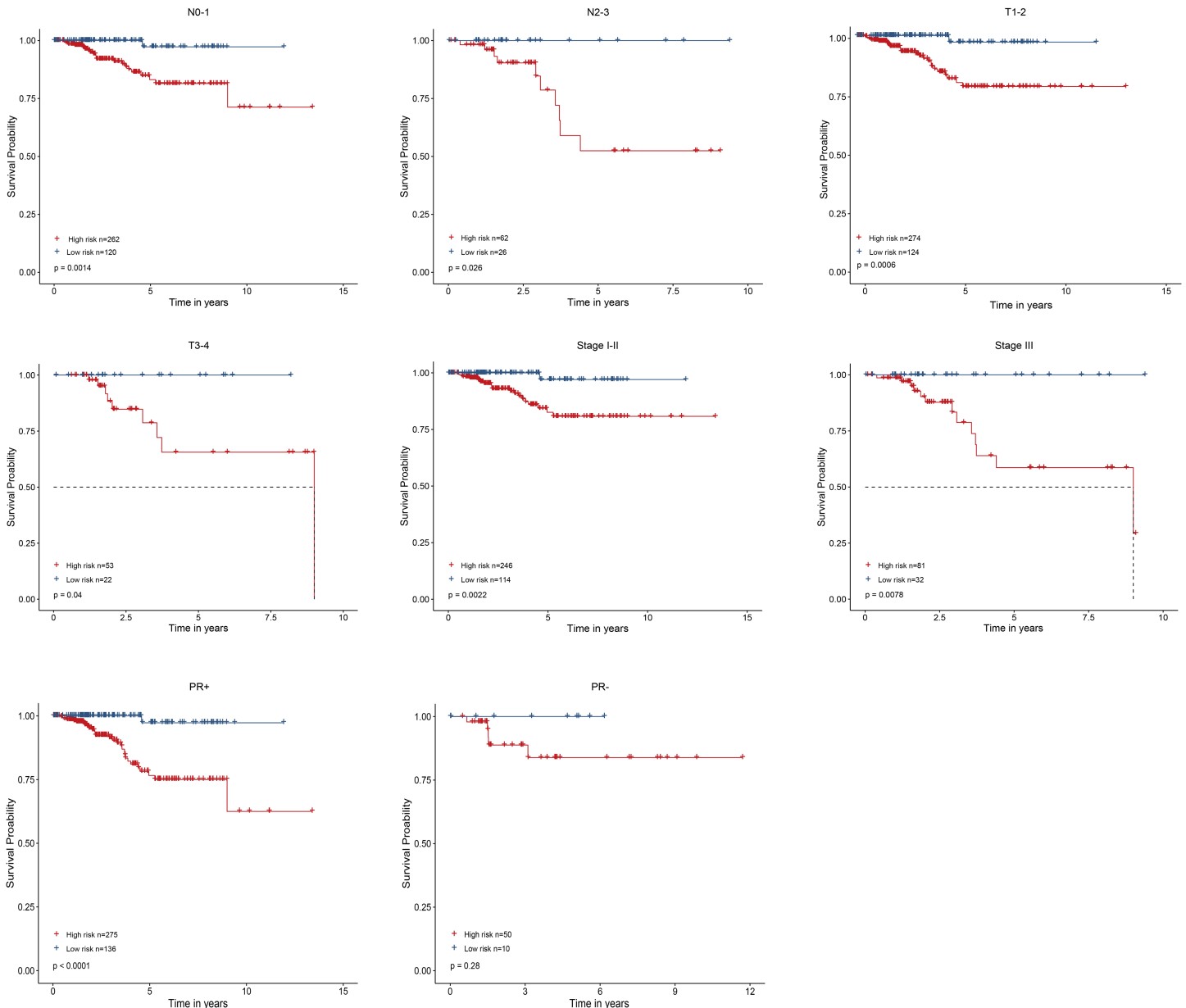

**Figure 4   Performed the risk stratified subgroup analysis according to clinical factors.**

multivariate Cox regression analyses filtering, the riskScore, T and Stage remained in the final Cox model for DFS (Figs. 7A–7B). A nomogram was established based on the final Cox model for DFS to evaluate of the 3- or 5-year DFS (Fig. 7C). The clinical stage and riskScore of the patient can be matched with this Nomogram to obtain the total points and estimate the survival rate of the patient. The nomogram achieved a C-index of 0. 740 (95% CI, 0. 653–0. 826), and the calibration plots demonstrated that the nomogram has good predictive power (Fig. 7D). Time-dependent ROC analyses also indicated that the nomogram was accurate in prediction of DFS compared to use riskScore or clinical risk

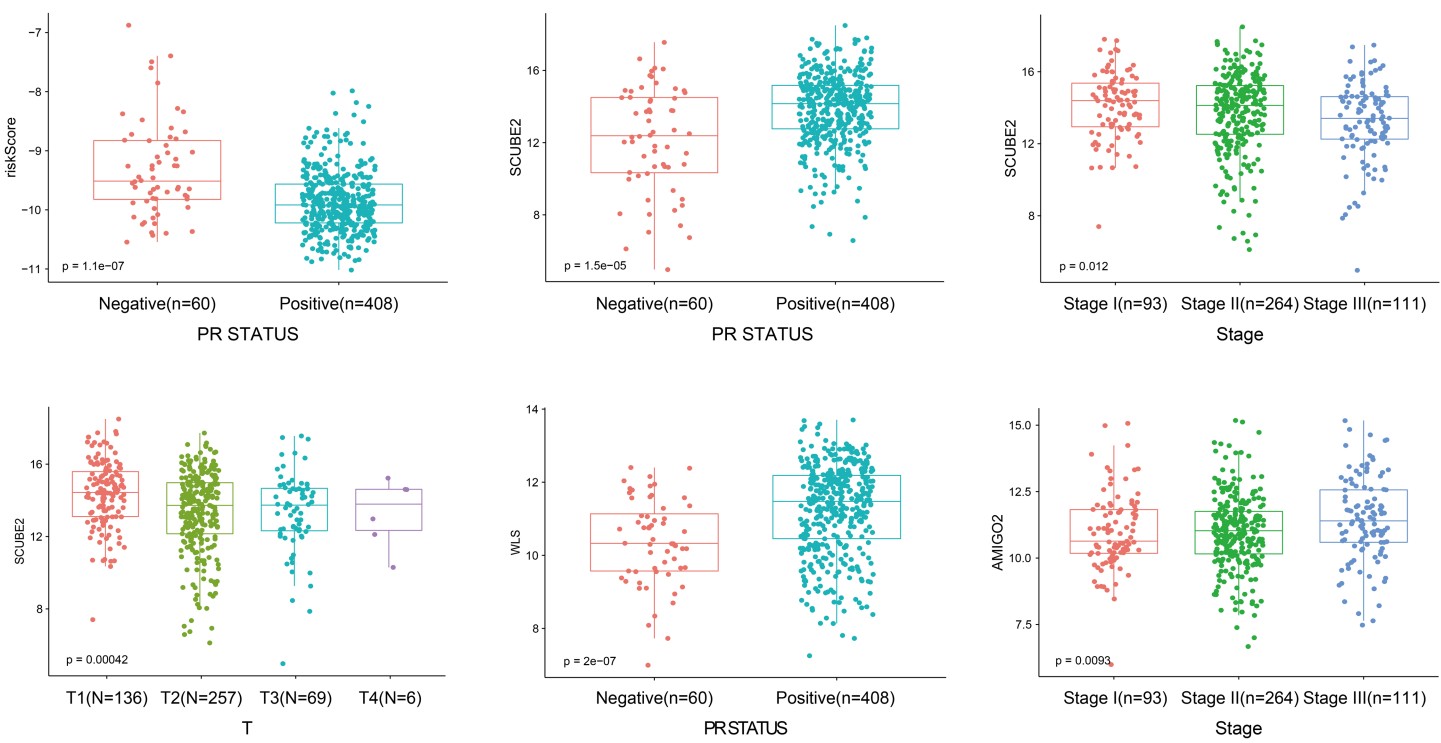

**Figure 5** **The expression of the four genes in different clinicopathological risk factors.**

factors alone (Fig. 7E). In conclusion, all the results suggest the clinical practical use of the nomogram for prediction of DFS.

## DISCUSSION

Breast cancer is highly heterogeneous in molecular and cellular, the morbidity of it increases every year. In recent years, molecular biomarkers of breast cancer become more and more important. Understanding the biological functions of breast cancer may offer new methods that can be used to predict and treat the disease. However, there are still few specific biomarkers with significant prognostic to predict survival of breast cancer. Therefore, in order to improve the prognosis of breast cancer, molecular screening of biomarkers is urgently needed.

In this study, we developed and validated a novel risk signature based on four resistance-related genes to predict patients with stage I-III ER+ and HER2- breast cancer. This article showed that our risk signature could effectively distinguish the breast cancer patients between low and high risk of survival groups. In addition, the four-gene based model was demonstrate to be independent from other clinicopathological features. Then, a comprehensive nomogram was constructed to predict the individualized DFS, which performed better than clinical factors and riskScore to classify the patients' survival risk. Finally, we explored the correlation between the four resistance-related DEGs and immune infiltrates.
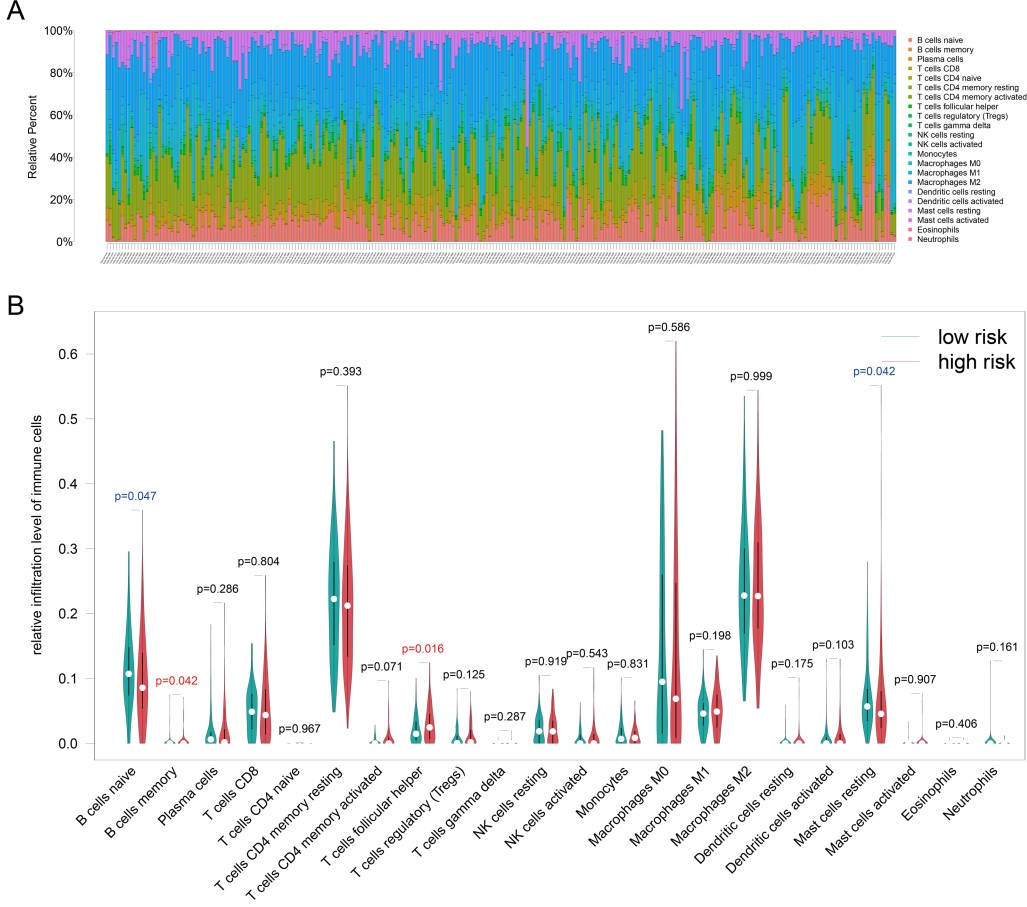

**Figure 6 Immune infiltration analyses results.** (A) The relative infiltration level of immune cells in pat ients of TCGA-BRCA dataset, the 22 immune cells were annotated by various colors under the legend. (B) T he difference immune cell infiltration levels between the high- and low groups.

Previous investigation proved that protein-based signature could predict tamoxifen treatment outcome in recurrent breast cancer and Long Noncoding RNA based signature could predict recurrence among ER+ breast cancer patients treated with tamoxifen (*De Marchi et al., 2016*; *Wang et al., 2018*). In reality, the expression profile-based prognostic biomarkers signature has been already investigated in several types of cancer, including glioblastoma multiforme, colon cancer, hepatocellular carcinoma, gastric cancer and non-small cell lung cancer (*Long et al., 2018*; *Qiu et al., 2020*; *Wang et al., 2020*; *Zuo et al., 2019*). But there is no research on the application of predictive models of resistance-related genes in ER+ and HER2- breast cancer.

There are four genes involved in building the risk signature in this study, they are *AMIGO2, LGALS3BP, SCUBE2* and *WLS*. All of these genes are protective markers in our study, there are no previous studies demonstrated their role in drug resistance. These genes paly different roles in different diseases.

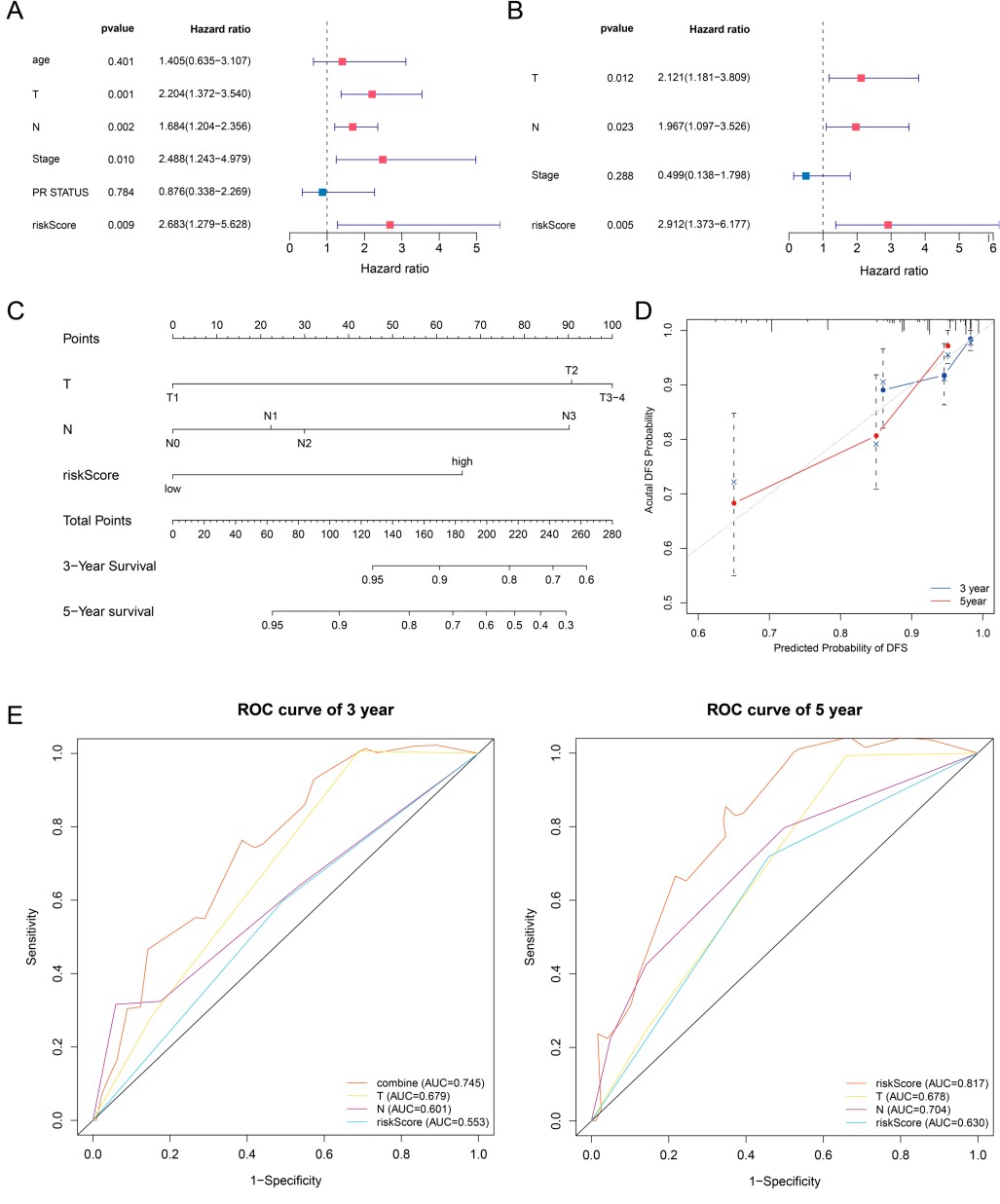

**Figure 7** **Building a prognostic nomogram combining riskScore with clinical risk factors.** (A,B) Univariate Cox (A) and multivariate Cox (B) association of the risk signa ture and clinicopathological characteristics with DFS in TCGA-BRCA. (C) Nomogram predicting 3- and 5-year DFS for patients with ER+ and HER2- breast cancer patients in stage I-III. (D) The calibration plots of the nomogram in prediction of the 3-year and 5-year DFS in the training cohort. (E) Time-dependent ROC analyses evaluate the accuracy of the nomogram and riskScore or clinical risk factors.

Adhesion molecule with IgG-like domain 2 (*AMIGO2*) is a multifunctional amphoterin-induced genes (*Kuja-Panula et al., 2003*). *AMIGO2* controls cell survival and angiogenesis in endothelial cells *via* Akt activation, knockout of *AMIGO2* can reduce cell survival and induce apoptosis (*Park et al., 2015*). *AMIGO2* also can promote liver metastasis by

regulating the tumor cell adhesion to liver endothelial cells and it function as a pro-proliferation factor in human melanoma *in vitro* and in vivo (*Fontanals-Cirera et al., 2017*; *Kanda et al., 2017*).

The lectin galactoside-binding soluble 3 binding protein (*LGALS3BP*), a large oligomeric glycoprotein, is overexpressed in many cancers and always associated with cancer growth and migration (*Woo et al., 2017*). *LGALS3BP* can induce vascular endothelial growth factor in human MDA-MB-231 cells and promotes angiogenesis, it also act as a E-selectin ligand that associated with metastasis and poor prognosis in ER- breast cancer (*Piccolo et al., 2013*; *Traini et al., 2014*). But in colon cancer, different from previous studies, *LGALS3BP* suppresses tumor proliferation in colon cancer cells, high expression of *LGALS3BP* have a good prognosis (*Piccolo et al., 2015*). There are no researches to explore the function of *LGALS3BP* in ER+ and HER2- invasive breast cancer.

The findings of Signal Peptide-CUB-EGF Domain-Containing Protein 2(*SCUBE2*) are controversial. *Chen et al. (2018b)* proved that upregulated *SCUBE2* expression in breast cancer stem cells can promote triple negative breast cancer aggression. But *Lin et al. (2014)* demonstrated that *SCUBE2* inhibited MDA-MB-231 cell proliferation. In MCF-7 breast cancer cell line, overexpression of *SCUBE2* protein suppresses proliferation, this result is consistent with our study (*Cheng et al., 2009*). *SCUBE2* also performed as protective factor in glioma, non-small cell lung cancer and colorectal cancer and inhibited the proliferation and migration (*Guo, Liu & Liu, 2017*; *Song et al., 2015*; *Yang, Miao & Li, 2018*).

The Wnt secretion protein Wntless (*WLS*) has been proved to be involved in the development of several human cancers. *WLS* mostly plays the role of oncogene in most tumors. In triple negative breast cancer, glioma and colon cancer, *WLS* promotes the cell proliferation (*Augustin et al., 2012*; *Voloshanenko et al., 2013*). In lung adenocarcinoma cell lines, WLS is upregulated by small nucleolar RNA host gene 17 (*SNHG17*), the up-regulation of WLS could reverse the inhibition of proliferation and promote apoptosis effects of *SNHG17* down-regulation (*Li et al., 2020b*). But in melanoma cell, *WLS* inhibits proliferation through the $\beta$-catenin signaling pathway and reduces spontaneous metastasis (*Yang et al., 2012*).

Tumor-infiltrating immune cells in the TME play a crucial role in antitumor and chemotherapeutic effects (*Denkert et al., 2010*; *Stanton & Disis, 2016*). Therefore, we used the CIBERSORT algorithm and the TIMER database to detect the relationship between risk signature and immune cells. The results proved that B cells memory and T cells follicular helper (Tfh) were positive correlated infiltrated in the high-risk group, but B cells naive and mast cells resting were negative correlated infiltrated in the high-risk group. Tfh cells are crucial for maintenance of humoral memory that assist B cells in producing germinal centers (*Crotty, 2019*). Mast cells may play protumoral or antitumoral roles in different tumor types, for instance, it could inhibit prostate adenocarcinoma development but promote the tumorigenesis of highly malignant neuroendocrine cancers (*Pittoni et al., 2011*) discovered.

This study also has some limitations. This is a retrospective research study and is susceptible to inherent biases. First, the study has potential selection bias and limited sample size. Then, the riskScore and nomogram cannot guide treatment strategies of breast

cancer, so that further verification and prospective trials are needed. Moreover, all of the data were obtained from the database, the population ethnicities in the TCGA database are mainly composed of White patients and Black patients; therefore; the findings of other ethnicities still require proof, and some other clinicopathological features including histological grade, are not contained in the nomogram. Because of the lack of clinical data in database, the nomogram has not been verified, so further researchs are needed to verify the nomogram. Finally, the precise mechanism of resistance-related genes involved in the risk signature need further *in vitro* and *in vivo* validation experiment.

## CONCLUSIONS

In this study, we used the data from TCGA-BRCA dataset and GEO database, constructed a risk signature based on four drug resistance-related genes and developed a nomogram to predict the survival of patients with stage I-III in ER+ and HER2- breast cancer. The results were validated in three GEO cohorts. All the results indicated that this risk signature might serve as novel molecular biomarkers for predicting breast cancer survival and provide guidance for therapeutic strategies.

### Funding
The authors received no funding for this work.

### Competing Interests
The authors declare there are no competing interests.

### Author Contributions

- Chen Shuai and Hongye He conceived and designed the experiments, analyzed the data, authored or reviewed drafts of the paper, and approved the final draft.
- Fengyan Yuan conceived and designed the experiments, analyzed the data, prepared figures and/or tables, and approved the final draft.
- Yu Liu performed the experiments, analyzed the data, prepared figures and/or tables, and approved the final draft.
- Chengchen Wang and Jiansong Wang performed the experiments, prepared figures and/or tables, and approved the final draft.

### Data Availability
This data is available at TCGA-BRCA and NCBI: GSE24460 and GSE67916.
https://portal.gdc.cancer.gov/

### Supplemental Information
Supplemental information for this article can be found online at http://dx.doi.org/10.7717/peerj.12202#supplemental-information.

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
