# Peer review of "Estrogen receptor—positive breast cancer survival prediction and analysis of resistance–related genes introduction"

_PeerJ, doi:10.7717/peerj.12202_

## Round 0.1 · original submission · Major Revisions

Please tightly follow the indications given by the referees. Discovery of new prognostic markers must also compare the markers identified with existing ones, in this case comparison at least with Recurrence score (Oncotype DX) and the intrinsic subtype signature (PAM50) is needed.

Reviewer 1 ·

Basic reporting

line 42-43. endocrine resistant -> endocrine resistance

line 63 The Cancer Genome Atlas

line 135 was -> were

line 148 I would remove the term “Regrettably”

Experimental design

I would have included to the supplementary materials an R script with all commands used to produce figures and analysis reported in this paper.

For example:

# dataset 1 has been downloaded from (link to the file), file name is "RNAseq.txt"

RNA.dataset <- read.table("RNAseq.txt")

# now make plot of PCA

library(packageX)

png(plot1.png)
plotPackageX(RNA.dataset, makePCA)
dev.off()

Validity of the findings

There are several critical points in this paper.

The first is related to model quality, in line 157 authors specify that AUC was between 0.686 and 0.714, suggesting that the four-gene risk signature had high sensitivity and specificity. In principle a value of AUC over 0.7 is considered acceptable [1], while a lower value is computed in most of the datasets used in this work (figure 3) . The AUC lower than 0.5 for single genes shows that their performance is a little worse than random, hence it’s difficult to support a method based on these predictors, even if some mathematical transformation is performed on them.

For multivariate cox analysis (figure 2 B), only WLS has a p-value < 0.5 and they all show a protective effect (according to the Hazard ratio) that doesn’t make sense at a first glance, maybe it should be better explained.

Figure 2, panel D, doesn't show a clear distinction between "Recurred/Progressed" and "DiseaseFree" patients.


In line 147 authors claim that WLS and SCUBE2 are protective factors. This is strange, it doesn’t fit with what is reported in literature, for example [2]


References

[1] Mandrekar JN. Receiver operating characteristic curve in diagnostic test assessment. J Thorac Oncol. 2010 Sep;5(9):1315-6. doi: 10.1097/JTO.0b013e3181ec173d. PMID: 20736804.

[2] Stewart, J., James, J., McCluggage, G. et al. Analysis of wntless (WLS) expression in gastric, ovarian, and breast cancers reveals a strong association with HER2 overexpression. Mod Pathol 28, 428–436 (2015). https://doi.org/10.1038/modpathol.2014.114

Reviewer 2 ·

Basic reporting

The title, abstract, introduction, methods, results and discussion are appropriate for the content of the text. Furthermore, the article is well constructed, the experiments are well conducted, and analysis is well performed. The figures are relevant, high quality, well labelled and described.

Experimental design

The experimental design is original and the research is within the scope of the journal. Research question is well defined, relevant and meaningful. The methods are highly technical, ethical and logistical.

Validity of the findings

All underlying data have been provided in detail. The findings are meaningful. The conclusions are well stated and relevant to the research questions.

Additional comments

This paper investigates the reliable prognostic signatures for ER+ and HER2- breast cancer
using TCGA dataset as discovery dataset, and GEO dataset as validation dataset. The authors identified 4 drug resistance-related genes as potential tools for ER+ and HER2- breast cancer risk stratification by developing a nomogram to predict the overall survival. Furthermore, the authors demonstrated the strong correlation between immune infiltration and the expression of these 4 genes or risk groups.

Editorial Criteria
BASIC REPORTING
The title, abstract, introduction, methods, results and discussion are appropriate for the content of the text. Furthermore, the article is well constructed, the experiments are well conducted, and analysis is well performed. The figures are relevant, high quality, well labelled and described.
EXPERIMENTAL DESIGN
The experimental design is original and the research is within the scope of the journal. Research question is well defined, relevant and meaningful. The methods are highly technical, ethical and logistical.
VALIDITY OF THE FINDINGS
All underlying data have been provided in detail. The findings are meaningful. The conclusions are well stated and relevant to the research questions.

Overall, I think this paper is novel and will be of interest to the community of breast cancer genetics, especially ER+ and HER2- breast cancer. The statistical part is valid and makes sense. The authors make it comprehensive by integrating analysis of multiple sources including GEO, TCGA. The main strengths of this paper is that it addresses an interesting and timely question, finds a novel solution based on a carefully selected set of rules and filters. As such this article represents an excellent and elegant bioinformatics study. Some of the weaknesses are the lack of in vitro or in vivo validation experiments. In general, the work is convincing except some major and minor comments below:


Major Comments:

I’m wondering if there are any ongoing clinical trials focusing on the 4 genes identified in this study in breast cancer? It will be very strong evidence for the significance of the current study if so.

I strongly recommend the authors to incorporate more data into their discovery datasets. The authors may also check if there are some consortiums focusing on breast cancer. I know a famous breast cancer dataset hosted in EGA: METABRIC (Molecular Taxonomy of Breast Cancer International Consortium https://ega-archive.org/studies/EGAS00000000083 ).

The Results of the Abstract didn’t mention the strong correlation between immune infiltration and the expression of these 4 genes or risk groups. However, it was mentioned in Methods in Abstract. I would recommend to add the results in the results as well.





Minor Comments:
It is great that a session of abbreviations was there to list all the abbreviations for the database names. I would recommend to also include abbreviations like IDC, ILC, TIL, DFS etc in that list.

All the gene names should be italic for all the gene names.

Figure 6 typo: replace “t he 22 immune cells” with “ the 22 immune cells”.


Paragraph from line 290: please include the lack of in vitro and in vivo validation experiment as one of the limitations.

Figure 1: replace “37DEGs” with “37 DEGs”.

Annotated reviews are not available for download in order to protect the identity of reviewers who chose to remain anonymous.

·

Basic reporting

1. Figure legends for the supplementary figures are missing.

2. Line98-99, how were the coefficients for each gene computed? Providing more information for clarity. Also elaborate on how a concordance index is used to discriminate and calibrate nomograms in the methods (Line109-110). The global p value for the best model shown in Fig2b should also be reported.

3. In Figure3, the survival plot figures should include more description, for example, what is the median survival for high risk and low risk?

4. Line113-114, and Line187, what is specifically the statistical test used to for the pathway analysis? Is it over-representation analysis or gene set enrichment analysis? Need to specify. Also it would be interesting to report what genes/gene sets are altered in expression that led to the altered pathway activity. Were any of the four genes in the signature in any of these significant pathways?

5. The figure5 needs to show the statistical test used in each case and the number of samples in each group.

6. How were the columns of Fig6A ordered? Was it ordered randomly or based on the low/high risk group?

7. The author should elaborate on the interpretation of nomogram and how it can be used to predict the survival of patients.

8. The r packages used for the analysis should also be cited, such as "rms", "clusterProfiler", "corrplot", etc.

Experimental design

1. In this study the authors did a very nice study of breast cancer prognostic biomarker discovery. They first identified genes that are differentially expressed between resistant cell lines and parental cell lines and then they established a link between these genes and survival probability of patients who are ER+ and HER2- . They demonstrated the independent prognostic value of the four-gene based signature and constructed comprehensive models to predict patients DFS. Very well done overall.

Validity of the findings

1. For the results presented in Fig6 B, since there are multiple tests for significance, are the P values reported after correction for multiple-testing? It is not mentioned in the body of the text nor legend.

2. In the multivariate models Fig7 b and Fig7 C, the relative importance of each variable is different, can the authors explain the discrepancy? Also what is the statistical test used to calculate the p values?

Additional comments

1. Overall very nice work. Just need to add more description to the figure legends and methods section as suggested above, as well as adding legends for supplementary figures.

---

## Round 0.2 · accepted · Accept

The authors have adequately addressed the issues raised.

Reviewer 1 ·

Basic reporting

no comment

Experimental design

no comment

Validity of the findings

no comment

Additional comments

no comment